# Using Importance–Performance Analysis to Reveal Priorities for Multifunctional Landscape Optimization in Urban Parks

**Xiaomin Xiao** , **Qiaoru Ye** and **Xiaobin Dong** *

Faculty of Geographical Science, Beijing Normal University, Beijing 100875, China;
202331051063@mail.bnu.edu.cn (X.X.); 202231051079@mail.bnu.edu.cn (Q.Y.)
* Correspondence: xbdong@bnu.edu.cn

**Abstract:** In the context of urban renewal, residents have presented elevated expectations for the quality of urban parks, necessitating the optimization of parks' multifunctional landscapes. Transforming residents' preferences for landscape services into a prioritized index for multifunctional landscape renewal poses a significant challenge. This study addresses this research gap by integrating importance–performance analysis (IPA) with residents' perception evaluations of landscape services. We establish an index system to evaluate perceptions of urban park landscape services. By employing the importance–performance analysis framework, we identify landscape service types that exhibit high importance but low satisfaction levels, thereby establishing priorities for multifunctional landscape renewal. Using Guangzhou's urban parks as a case study, our findings reveal variations in users' demands for different landscape services and differences in demand among various user groups for similar services. Users assign utmost importance to safety services while expressing the highest satisfaction with physical and mental health or microclimate regulation services. Significant disparities exist between middle-aged/elderly groups and young people regarding perceptions of social interaction, waste disposal, and sense of belonging services. Our results demonstrate that IPA analysis can elucidate priorities for multifunctional landscape renewal, facilitate public participation in improving urban park landscapes, and provide decision-making support for optimizing these landscapes.

**Keywords:** multifunctional landscape; importance–performance analysis (IPA); landscape services; perception evaluation; urban parks

## 1. Introduction

In recent years, China and numerous other regions worldwide have witnessed a rapid process of urbanization, resulting in swift transformations in population structure, travel patterns, population distribution, lifestyle choices, landscape perception, and aesthetic preferences. These profound changes have significantly influenced human behavior and altered the dynamic between humans and land. Moreover, they have reconfigured the interplay between ecosystem services and human beings by highlighting the necessity to address diverse social groups' requirements for such services, including the elderly, children, new urban immigrants, as well as individuals left behind in rural areas [1]. Ecosystem services serve as the fundamental basis for multifunctional landscapes, and landscape practitioners should employ innovative approaches to comprehensively examine and understand the intricate relationship between ecosystem services and human needs in order to shape truly multifunctional landscapes [1].

Since the inception of the concept of "multifunctional landscape" at the International Symposium on "Multifunctional Landscape—Interdisciplinary Approaches to Landscape Research and Management" in 2000, it has gained widespread recognition in landscape analysis, evaluation, planning, and management research [2]. The notion of a multifunctional landscape encompasses a landscape that not only fulfills its primary ecological

function but also incorporates additional functions such as social, economic, cultural, historical, and aesthetic aspects. Moreover, it emphasizes the intricate interplay between these diverse functions [3]. This concept holds immense potential for bridging landscape practice with societal transformations by facilitating stakeholder coordination and offering novel perspectives to comprehend the intricate relationship between humans and nature while promoting sustainable development. The primary reason is that the cognition of multifunctional landscapes arises from human value judgments [4]. For instance, the quantitative assessment of landscape functions is significantly influenced by the emotions, cognitive abilities, and values of local residents [5]. Therefore, stakeholders are encouraged to actively engage in the decision-making process alongside relevant experts and planners for multifunctional landscape planning and management [6]. Currently, research on multifunctional landscapes primarily focuses on their development and transformation without considering stakeholder perspectives for optimizing their multifunctionality. This oversight may result in resource wastage or social conflicts [7,8].

Ecosystem services are fundamental components for the realization of a multifunctional landscape, while a multifunctional landscape represents the comprehensive trade-off outcomes of ecosystem services elevated to the level of landscape services; transitioning from ecosystem services to a multifunctional landscape essentially involves transforming from natural ecosystems (function and service) to human-centered considerations, taking into account nature [9]. The concept of landscape services emerges through an in-depth exploration of ecosystem services, with its core essence referring to the products or services provided by landscapes to humans that are closely linked to human well-being [10–14]. Landscape services emphasize the significance of spatial patterns and allocations of landscape elements, serving as interfaces for human–environment interactions that can be easily perceived by users [15]. Compared with ecosystem services, the concept of landscape services enables local stakeholders to directly engage in operating and managing multifunctional landscapes [16,17]. Currently, both domestic and international laws support the development of multifunctional urban park landscapes based on residents' needs. For instance, the Measures for the Administration of Urban Parks issued by China's Ministry of Housing and Urban-Rural Development propose transforming urban parks into shared green spaces that cater to diverse groups. Similarly, Guangzhou's Regulations on Parks advocate for public participation in park development to meet a wide range of requirements. However, despite these efforts, urban renewal practices in China predominantly rely on top-down decision making with limited community and resident consultation [18]. In contrast, the United States National Park Service Organic Act emphasizes public rights to reasonable use and enjoyment of natural resources. The Japanese Natural Park Law highlights the importance of incorporating community opinions into park planning decisions. Additionally, South Korea's Urban Park Law underscores the need for urban parks to address people's varied needs. In recent years, the United Kingdom has also recognized the significance of empowering communities and strengthening both top-down and bottom-up mechanisms in shaping new urban park policies [19].

Currently, there is limited research exploring the trade-off mechanism of human perception of landscape function and multifunctionality [9]. Existing studies on urban park perception primarily rely on social media big data [20–22]. For instance, Wang utilizes social media data to uncover variations in urban parks' provision of human welfare services [20] and quantifies visitors' perception and satisfaction towards Beijing's urban parks based on Dianping data [23], while Kong et al. employ social media big data to investigate the impact of urban parks on people's emotions [23]. These studies demonstrate that big data can be utilized for analyzing users' demand for landscapes; however, certain limitations exist. Social big data predominantly reflects the needs of young individuals and cannot adequately represent other demographic groups, such as the elderly [24,25]. Moreover, concerns arise regarding the reliability of big data in accurately reflecting user perception [26]. In summary, due to its lack of personal information about users, big data fails to facilitate effective user profile analysis and consequently hampers a comprehen-

sive understanding of visitors' perceptions toward urban parks [27]. Considering that factors like wealth, age, ethnicity/religion, and formal education influence users' roles and decision-making processes [28], current literature rarely addresses the contribution made by different demographic groups toward multifunctional landscapes.

However, optimizing multifunctional landscapes based on residents' demand preferences for landscape services poses several challenges. Firstly, there is a need to quantitatively evaluate landscape services from the perspective of users. Currently, quantitative evaluation methods for landscape services primarily rely on economic approaches such as the market price method for analyzing water supply value, tourism cost method for assessing recreational value, shadow engineering method and results reference method for evaluating cultural research value, and premium income method for analyzing human settlements improvement value [29]. However, these evaluations only consider the monetizable aspects of land and resources while often neglecting non-monetary forms of value such as social and cultural significance [30–33]. Therefore, it is essential to incorporate dimensions of social value or non-economic valuation that encompass intrinsic human worth [34–36]. The focus of this study lies in quantitatively evaluating landscape services from the perspective of users and optimizing multifunctional landscapes. Secondly, how can we identify residents' demand preferences for landscape services? Currently, the application of landscape services mainly focuses on large-scale land use optimization [37], biodiversity conservation [38], and landscape planning to address climate change [39]. This approach is mostly based on the potential supply of ecosystem services without considering human demand for landscape services [40]. On a small scale, cultural landscape services are primarily used to meet residents' leisure needs with less consideration given to habitat function, production function, and other adjustment functions. Furthermore, as it is difficult to characterize and quantify ecosystem services at the landscape scale [38], there is an imbalance between the supply and demand of landscape services. Therefore, in multifunctional landscape planning applications, the key challenge lies in applying the concept of landscape services during spatial planning processes while identifying relationships between the supply and demand for these types of service offerings [40]. This study aims to quantitatively analyze residents' demand preferences for various types of park landscapes from their perspective. Finally, how can the translation of residents' demand preferences for landscape services into the prioritization of multifunctional landscape renewal be achieved? In addition to considering stakeholders and conducting quantitative assessments of landscape services, optimizing multifunctional landscapes also necessitates balancing priorities to enhance optimization benefits and achieve the optimal combination of multiple public benefits [41]. However, determining which landscape services should be improved first remains a persistent challenge [42].

In order to address the research gap, this study aims to develop a comprehensive landscape optimization model using the importance–performance analysis (IPA) approach, with the objective of conducting an in-depth analysis of subtle variations in individuals' demand for landscape services and determining priorities for enhancing multifunctional landscapes. The IPA method facilitates the efficient allocation of limited resources by prioritizing attributes that require improvement [43–45]. Originally designed for market research to assess consumer satisfaction and prioritize supply strategies accordingly [46], IPA can effectively optimize multifunctional landscapes, enabling urban landscapes and related decisions to strike a balance between public goals and personal interests while reconciling conflicting interests [47–50]. IPA and its extended versions have found wide applications across various domains, including tourism [51,52], education [53], and medical and health sectors [54].

In recent years, IPA has been increasingly utilized in evaluating the urban environment to enhance environmental quality and services. The extensive application of IPA in urban environment renewal underscores its potential [42]. For instance, Arijit Das et al. employed the IPA model to assess the degradation status of the Chatra Wetland in India and proposed optimization suggestions [55]. However, this study only considered 10 commonly recog-

nized ecosystem services, which imposes certain limitations on wetland restoration efforts. Keith and Boley utilized the IPA model to analyze residents' perspectives on the physical characteristics of the greenway along Atlanta's Ring Road and subsequently optimized its design based on these findings [56]. Swapan et al. employed IPA to compare the user evaluation of ecosystem services between Dufu Cottage in Chengdu and King's Park in Perth [57]. Ou et al. utilized IPA to assess the importance and satisfaction of residents regarding urban park soundscapes, providing guidance for planning and design [58]. However, their study solely focused on soundscapes and was limited to shaping multifunctional landscapes. Shijie Gai et al. applied the IPA model to analyze the significance and performance of residents' perspectives on cultural ecosystem services provided by Beijing parks, considering social group differences, and proposed suggestions for optimizing park cultural system services [59]. Zheng et al. assessed the sensory satisfaction of typical urban parks in Beijing using the IPA model [60], which compared satisfaction levels across different senses and provided recommendations for park optimization. Diverging from previous studies, this research endeavors to explore perceptual differences and user group disparities by evaluating multiple types of ecosystem services (beyond cultural ecosystem services) and employing the IPA analysis method for prioritizing multifunctional landscape optimization. Consequently, it broadens the application scope of IPA while holding both theoretical and practical significance in shaping multifunctional landscapes.

To address the knowledge gap in multifunctional landscape optimization from the perspective of park users' perception and evaluation of landscape services, this study establishes a user-centric IPA-based model for optimizing multifunctional landscapes in urban parks. By utilizing landscape service perception as the evaluation index, this model efficiently identifies optimization priorities and allocates resources to construct scientifically designed multifunctional landscapes. The main issues addressed include the following: (1) quantitatively analyzing users' preferences and satisfaction levels regarding various landscape services in urban parks; (2) investigating potential differences in perception among different populations towards different landscape services and optimizing multifunctional landscapes based on diverse population needs; and (3) employing the IPA method to transform importance and satisfaction evaluations into prioritized actions for multifunctional landscape optimization.

## 2. Materials and Methods

### 2.1. Study Area Overview

Urban parks play a crucial role as green spaces within cities, offering comprehensive ecosystem services to meet the physical and mental rejuvenation needs of citizens. Since the 1960s, there has been an exponential increase in demand for and utilization of landscape services [10]. Relevant research indicates that individuals residing in natural environments experience enhanced well-being [61]; proximity to green spaces effectively alleviates psychological stress [62], and residents living near such areas exhibit a heightened sense of community and social safety [63].

This study utilizes urban parks in Guangzhou as a means to optimize the multifunctional landscape. As an exemplar of China's megacities, Guangzhou is confronted with social and environmental challenges arising from rapid urbanization, such as the disparity between the supply and demand of urban public space. Urban parks serve as crucial public spaces that offer diverse ecosystem services for citizens, with their ecological benefits serving as significant indicators in Chinese urban planning [64]. In accordance with the Special Planning for Construction and Preservation of Parks in Guangzhou (2017–2035), the city aims to establish 828 parks, including 800 community parks and street parks, thereby enhancing the urban landscape and improving residents' quality of life.

The selected research objects include Haizhu Lake Park, Pearl River Park, and Yunxi Ecological Park (Figure 1, Table 1). These parks were chosen for the following reasons: (1) They are comprehensive ecological urban parks that offer a diverse range of landscape services. (2) Situated in densely populated areas, they serve as significant locations for citi-

zens to experience urban park landscapes. (3) They hold a crucial position in Guangzhou's green space planning system, representing an opportunity to enhance the provision of urban park landscape services and optimize multifunctional landscapes.

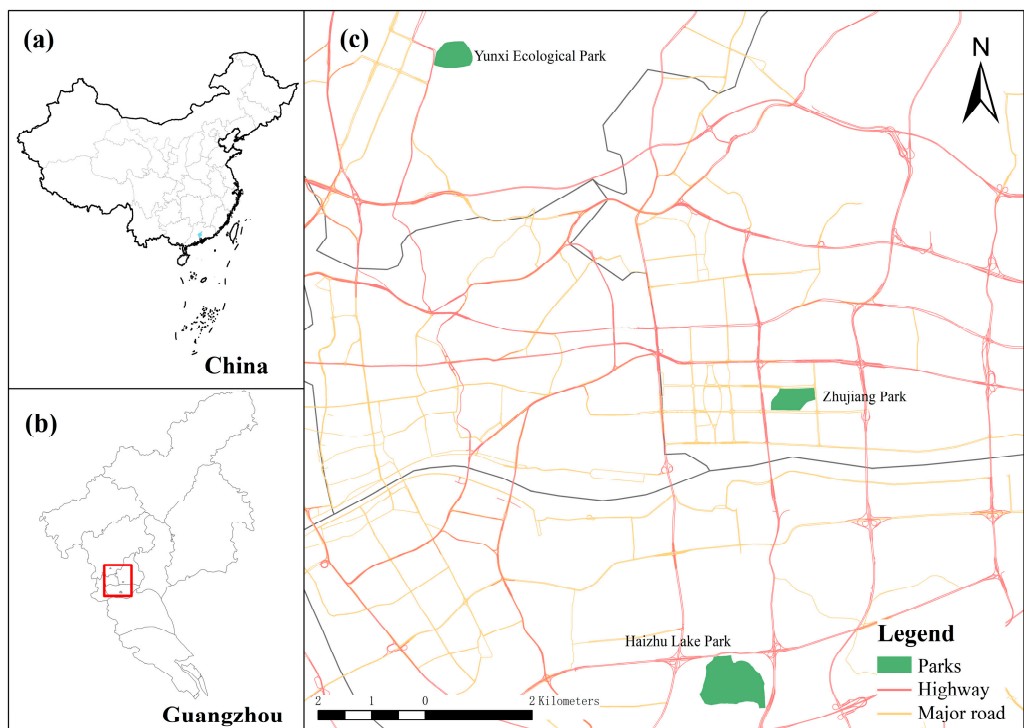

**Figure 1.** Locations of studied parks. (**a**) illustrates the geographical location of Guangzhou within China, (**b**) depicts the specific positioning of (**c**) within Guangzhou, and (**c**) showcases the precise location of studied parks.

**Table 1.** Overview of parks.

| Park | Completion Date | Location | Area | Functional Positioning | Main Attractions |
|---|---|---|---|---|---|
| Zhujiang Park | 2000 | Pearl River New City Center hinterland, Tianhe District, Guangzhou | The total area is 28 hectares, water area is 3.64 hectares, and land accounts for 13%. | An ecological park integrating culture, viewing, leisure, and recreation | Shady Plantation Garden, Aquatic Plantation Garden, Fast Green Plantation Garden |
| Haizhu Lake Park | 2011 | Haizhu District, Guangzhou City | The total area is 22.48 hectares, with 7.95 hectares of water area, accounting for 35%. | An ecological park for rainwater storage, recreation, viewing, and culture | Entrance Square, Green Island, Mission Hills, and Riverside |
| Yunxi Ecological Park | 2001 | South of Baiyun Mountain Scenic Area, Guangzhou | The total area is 17.56 hectares, and the green area is 15.83 hectares, accounting for 90.16%. | Ecological park for recreation and enjoyment | Water-stacking Garden, Fruit Fragrance Garden, Lotus Garden |

*2.2. Method*

2.2.1. Revising the Indices for Evaluating Landscape Service Perception

The optimization of multifunctional landscapes necessitates the selection of appropriate landscape service indicators, as these indicators serve as a crucial platform for elucidating key issues and priorities [65,66]. The classification of landscape services should be both objective and scientific while also reflecting human beings' subjective perceptions and needs. Furthermore, it is essential to differentiate between the functions and outcomes of landscape services in order to avoid redundant calculations. De Groot et al. [13] categorized landscape services into supply services, regulation services, habitat services, and cultural services; Hermann et al. [67] divided them into supply services, regulation services, habitat services, information services, and bearing services; and Valles-Planells et al. [68] classified them as supply services, regulation and maintenance services, and socio-cultural services. Based on existing literature and research requirements, this paper adopts De Groot's classification type for landscapes while implementing the following method: (1) consulting domestic and foreign literature on ecosystem services along with referencing the selection of ecosystem service indicators; (2) selecting landscape service evaluation indicators that are suitable for the actual situation in three urban parks. According to previous landscape index systems proposed by scholars such as Mandr et al. [31] and Zhang et al. [69], certain services related to land fertility maintenance, erosion control, and biological control were excluded from the regulation service due to their inconsistency with the functions of urban parks and residents' perceptions. However, considering that urban parks are public spaces with distinct regional characteristics, cultural services were expanded to include security, sense of place, and social integration aspects. Additionally, food supply was incorporated into the supply service category due to the common practice of planting fruit trees in Guangzhou parks. Consequently, a set of 19 detection indices was formulated (refer to Table 2).

**Table 2.** Evaluation indices of urban park landscape service.

| Landscape Service Type | Landscape Service Index | Literature Source |
|---|---|---|
| Supplying services | Food production | Gretchen Daily [70]; Costanza et al. [71]; MEA [10]; Buchel et al. [72] |
| | Raw material supply | Gretchen Daily [70]; Costanza et al. [71]; MEA [10]; Buchel et al. [72] |
| Adjusting services | Air purification | Gretchen Daily [70]; Costanza et al. [71]; Bolund et al. [47]; MEA [10]; Buchel et al. [72]; De Groot [73] |
| | Microclimate regulation | Gretchen Daily [70]; Costanza et al. [71]; Bolund et al. [47]; OUYANG Zhiyun et al. [74]; MEA [10]; Buchel et al. [72]; De Groot [73] |
| | Rainwater storage | Gretchen Daily [70]; Costanza et al. [71]; Bolund et al. [47]; OUYANG Zhiyun et al. [74]; MEA [10]; Buchel et al. [72]; Ungaro et al. [75] |
| | Waste disposal | Gretchen Daily [70] |
| | Disaster adjustment | Gretchen Daily [70]; Costanza et al. [71]; OUYANG Zhiyun et al. [74]; MEA [10]; Buchel et al. [72] |
| | Noise mitigation | Bolund et al. [47]; Buchel et al. [72] |
| Habitat services | Biodiversity | Gretchen Daily [70]; Costanza et al. [71]; OUYANG Zhiyun et al. [74]; MEA [10]; Buchel et al. [72]; Ungaro et al. [75]; Rall et al. [76] |
| | Nutrient cycling | Gretchen Daily [70]; Costanza et al. [71]; OUYANG Zhiyun et al. [74]; MEA [10]; Buchel et al. [72] |

**Table 2.** *Cont.*

| Landscape Service Type | Landscape Service Index | Literature Source |
|---|---|---|
| Cultural services | Inspiration | Gretchen Daily [70]; MEA [10]; Buchel et al. [72]; Swapan et al. [57]; De Groot [73]; Ungaro et al. [75]; Rall et al. [76]; Canedoli et al. [77] |
| | Educational value | Gretchen Daily [70]; MEA [10]; Buchel et al. [72]; Swapan et al. [57]; Rall et al. [76]; Langemeyer et al. [78] |
| | Recreation and entertainment | Gretchen Daily [70]; Costanza et al. [71]; Bolund et al. [47]; MEA [10]; Buchel et al. [72]; Swapan et al. [57]; De Groot [73]; Speak et al. [79]; Langemeyer et al. [78] |
| | Physical and mental health | MEA [10]; Brunner [62]; Mitchell et al. [80]; Buchel et al. [72] |
| | Cultural preservation | Gretchen Daily [70]; MEA [10]; Swapan et al. [57] |
| | Spiritual value | MEA [10] |
| | Social interaction | MEA [10]; Kuo [63]; Buchel et al. [72]; Langemeyer et al. [78]; Speak et al. [80]; Rall et al. [76] |
| | Sense of belonging | Kuo [63]; MEA [10]; Buchel et al. [72]; Langemeyer et al. [78]; Rall et al. [76] |
| | Safety | MEA [10] |

2.2.2. Quantifying Residents' Evaluation of Landscape Services

Firstly, the importance and satisfaction of 19 landscape service indicators were assessed using a 5-point Likert scale. Importance was rated on a scale of 1 to 5, with 1 being "very unimportant" and 5 being "very important". Satisfaction was also rated on a scale of 1 to 5, with 1 being "very unsatisfied" and 5 being "very satisfied". Additionally, user opinions regarding landscape services were gathered through questionnaires designed to evaluate the importance and performance of each service. The importance and satisfaction scores were then standardized.

Secondly, the reliability and validity test is conducted to verify the internal consistency of the evaluation factors and the reliability of factor analysis, which serves as a fundamental step for subsequent statistical analyses. This study employs internal reliability analysis, specifically Cronbach's alpha coefficient test. In Cronbach's alpha analysis, a reliability coefficient higher than 0.8 indicates high reliability; between 0.7 and 0.8 suggests good reliability; between 0.6 and 0.7 implies acceptable reliability; and below 0.6 signifies poor reliability. Additionally, this study utilizes content validity analysis where effective sample data undergoes KMO sampling appropriateness test and Bartlett sphericity test to assess its suitability for factor analysis. A higher KMO value approaching unity reflects stronger variable correlations and greater suitability for factor analysis purposes. When KMO > 0.5, the data are deemed suitable for factor analysis.

2.2.3. Analysis of Residents' Perception Differences on Landscape Services

Firstly, a descriptive statistical analysis was conducted using SPSS 25.0 to calculate the mean and standard deviation (representing data variability) of each landscape service index. The average score allows the identification of urban residents' emphasis and satisfaction levels for each landscape service while also distinguishing differences in their emphasis and satisfaction degrees. Furthermore, the standard deviation is utilized to analyze subtle variations between the importance and performance of different landscape services [81].

Furthermore, the paired-samples *t*-test is employed to examine the statistical significance of discrepancies between the mean values of two correlated samples and their overall representatives. In this study, this approach is utilized to assess disparities in importance and satisfaction levels pertaining to landscape service indicators.

2.2.4. Prioritization of Multifunctional Landscape Optimization through IPA Analysis

The IPA analysis method assesses the current state of users' perception of landscape services in urban parks based on their importance and satisfaction levels. The IPA analysis model is constructed with importance as the X-axis and satisfaction as the Y-axis (as depicted in Figure 2). The dotted line represents the average score for importance and satisfaction [51,82]. Four quadrants are distinguished: the first quadrant corresponds to the advantageous area, which signifies key attributes that should be preserved and maintained; the second quadrant represents the maintenance area, indicating attributes that are advantageous and should remain unchanged to meet existing user demands; the third quadrant denotes the opportunity area, encompassing priority areas for improvement; finally, the fourth quadrant signifies the improvement area where users attach great importance but reasonable demands have not been met by suppliers. This zone requires enhancements. The roles of the IPA analysis method include the following: (1) quantitatively analyzing users' perceived importance of landscape services in urban parks along with identifying gaps between perception and reality, and (2) analyzing coordinate point distribution within four quadrants to determine priority areas for multifunctional landscape optimization while scientifically proposing strategies for such optimization.

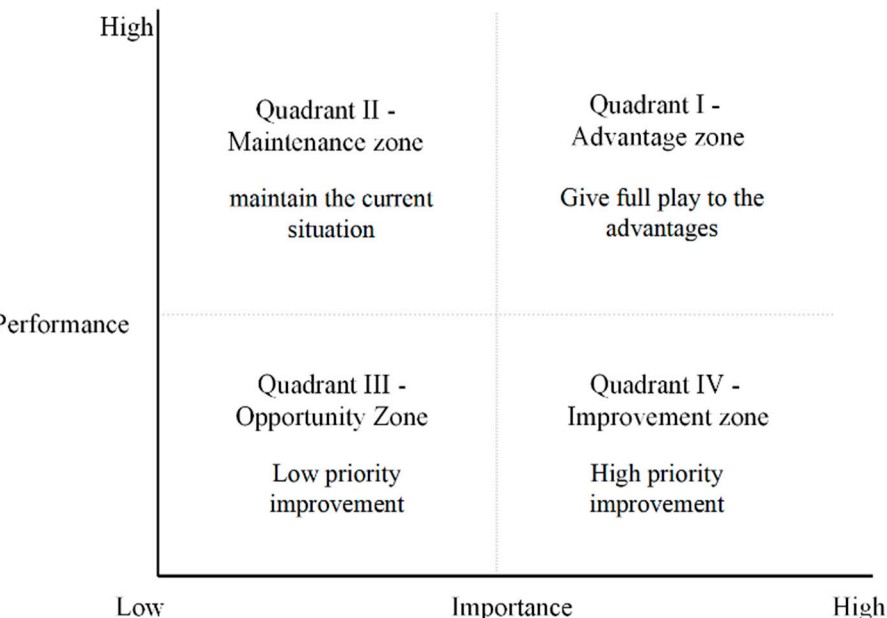

**Figure 2.** Concept map of importance and performance.

2.2.5. Identifying Different Groups of People and Their Preferences for Multifunctional Landscapes

Firstly, the second-order clustering analysis was conducted using SPSS 25.0 to unveil latent groupings or clusters within the dataset based on multiple indicators, which may not be readily apparent otherwise. Consequently, employing this method can facilitate the categorization of distinct user types. Secondly, by integrating IPA analysis, we further examined the demand preferences of different user groups for urban park landscape services to provide more targeted recommendations for optimizing multifunctional landscapes in urban parks. For detailed research methods, research contents and research framework Refer to Figure 3.

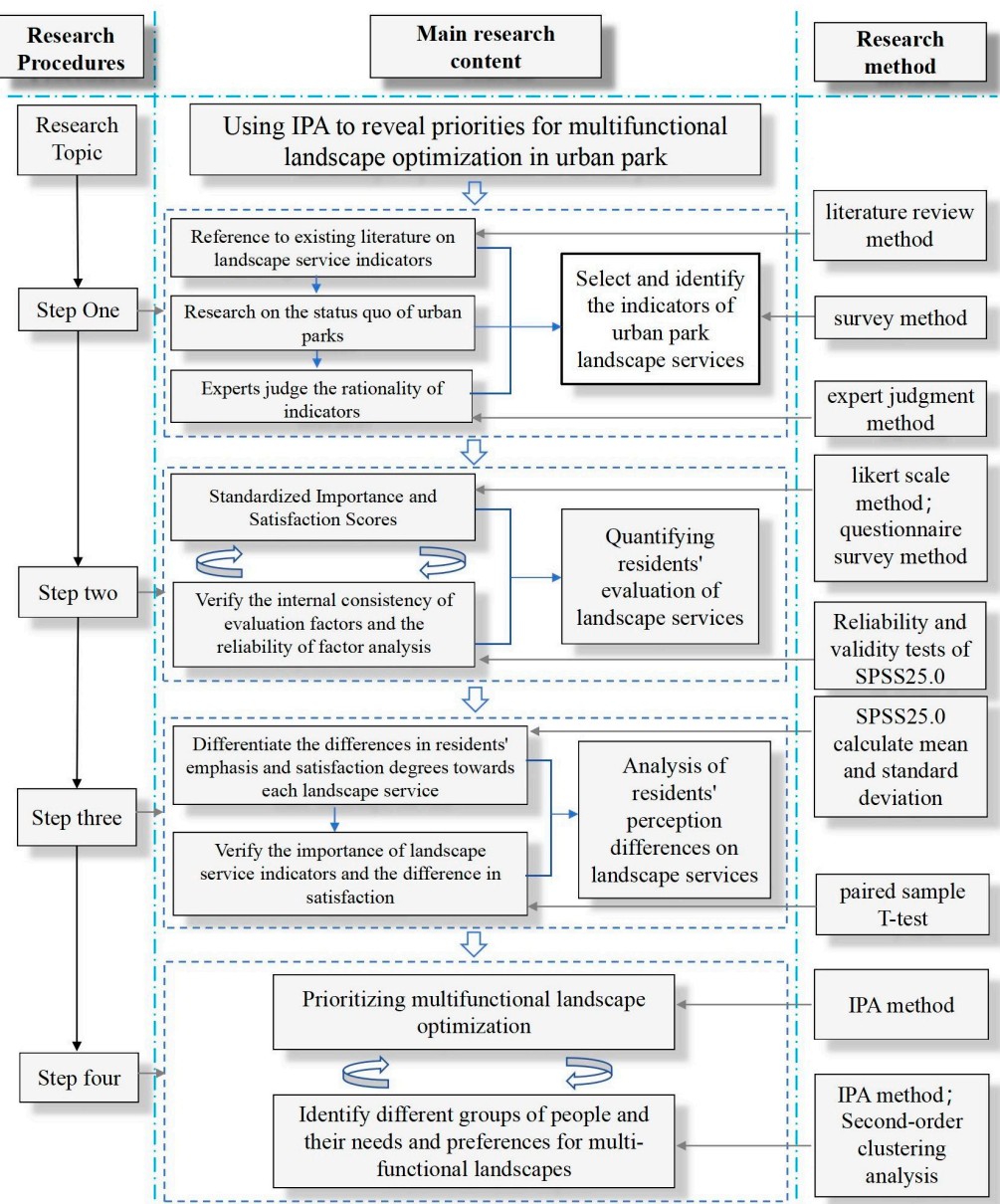

**Figure 3.** Framework of this study.

*2.3. Data Acquisition and Analysis*

The designed questionnaire on the utilization of urban parks in Guangzhou comprises two main sections. The first section encompasses respondents' basic information and their park usage, including age, gender, occupation, education level, duration of residence in Guangzhou, family income range, frequency of park visits, and duration spent in the park. The second section focuses on respondents' evaluation of the significance and satisfaction regarding landscape service assessment indicators within the park. The measurement method is Likert's five-point integral scale method. To ensure comprehensibility for ordinary citizens who may not be familiar with academic terminology related to landscape service indices, questions in the questionnaire have been transformed into colloquial language based on these indices. For instance, "safety" is rephrased as "I feel secure when visiting this park", while "cultural preservation" is modified to "I can explore cultural heritage sites and historical landmarks within this park" (Table 3). This approach aims to employ importance and satisfaction as criteria for screening landscape service evaluation indicators from a citizen's perspective.

**Table 3.** Perception of landscape services in urban parks.

| Landscape Service Type | Landscape Service Index | Perception of Landscape Service |
|---|---|---|
| Supplying services | Food production | You can pick some fruits and other ingredients or food. |
| | Raw material supply | A cultivated area for some medicinal herbs, flowers, and seedlings. |
| Adjusting services | Air purification | Purify the air and make the air fresher. |
| | Microclimate regulation | You feel the climate is more comfortable and pleasant. |
| | Rainwater storage | The park can absorb, store, and purify rainwater. |
| | Waste disposal | The park can realize the degradation and absorption of waste. |
| | Disaster adjustment | The park can alleviate natural disasters such as floods, sandstorms, and smog. |
| | Noise mitigation | You feel the environment is quiet and not noisy. |
| Habitat services | Biodiversity | There are a wide variety of plants and animals. |
| | Nutrient cycling | The oxygen levels are high and the carbon dioxide levels are low during the day in the park. |
| Cultural services | Inspiration | You can enjoy the beautiful scenery and receive some inspiration. |
| | Educational value | The park has educational significance. |
| | Recreation and entertainment | An important place for exercise, entertainment, leisure, and other activities. |
| | Physical and mental health | You can be close to nature, relax the body and mind, and relieve stress. |
| | Cultural preservation | You can visit some cultural heritage sites and historic sites. |
| | Spiritual value | You are able to burn incense and attend temple fairs. |
| | Social interaction | You are able to socialize and gather with friends and family. |
| | Sense of belonging | It feels like home. |
| | Safety | You feel safe. |

This questionnaire primarily targets users who have visited Haizhu Lake Park, Pearl River Park, and Yunxi Ecological Park. The survey period spans from 18 August to 24 August 2019. The "Guangzhou City Park Usage Questionnaire" is disseminated among Guangzhou citizens through the "Questionnaire Star" App function. A total of 542 valid questionnaires were collected, with 209 obtained from Haizhu Lake Park, 188 from Zhujiang Park, and 145 from Yunxi Ecological Park.

2.3.1. Demographic Characteristics of the Respondents

Through the analysis of the demographic characteristics of the respondents (Table S1), it is observed that among the 542 respondents, there was an almost equal proportion of male (52.2%) and female (47.8%) participants, indicating that the three parks possess a certain allure for both genders. This suggests that urban parks should incorporate service functions that cater to the needs of both men and women. In terms of occupation, freelancers constituted the largest group of respondents at 22.7%, followed by individuals from the commercial service industry, students, and retirees, accounting for a combined total of 39.3%. Notably, retirees comprised 30.3% of respondents in Yunxi Ecological Park, whereas farmers, soldiers, workers, party and government officials, employees of enterprises and institutions, as well as unemployed individuals were underrepresented among park-goers. In terms of education, the majority of respondents hold postgraduate degrees or below, constituting 94.3% of the sample. Regarding annual family income, a significant proportion (66.8%) reported an income below CNY 140,000 per annum. With respect to age distribution, a predominant percentage (97%) comprises young and middle-aged individuals; notably, Yunxi Ecological Park attracts more middle-aged and elderly visitors. Furthermore, 62.2%

of participants visit the park at least once a week, while 37.8% go on a monthly or sporadic basis. Considering the duration spent in the park, less than four hours is sufficient for 95.2% of respondents. From the perspective of the duration of respondents' stay in Guangzhou, it is evident that 88.7% are long-term residents, while only 7.7% are tourists and 3.6% are recent inhabitants. This indicates that the primary beneficiaries of the three city parks over an extended period are long-time citizens who have integrated these parks into their daily lives. In order to cater to their needs effectively, future enhancements in ecological services should be considered. Furthermore, analysis of age demographics reveals that young and middle-aged individuals constitute a majority among the respondents.

2.3.2. Reliability and Validity Test of Landscape Service Evaluation Index

The reliability (Table S2) and validity (Table S3) of the park data were thoroughly examined and analyzed. The Cronbach's alpha values for the importance and satisfaction indices of all three parks exceeded 0.8, indicating a high level of reliability and stability in the data. Additionally, the KMO values for both satisfaction and importance indices related to landscape services across all three parks surpassed 0.8, suggesting excellent sample validity. Moreover, the Bartlett test revealed a significant correlation between variables with a significance probability (sig = 0.000 < 0.05). Consequently, further analysis can be conducted on this dataset.

## 3. Results

*3.1. Analyzing the Evaluation of Urban Park Landscape Services from Residents' Perspective*

3.1.1. Evaluation of the Importance and Satisfaction of Landscape Service in Urban Parks

The evaluation of the importance and satisfaction of 19 landscape services in Haizhu Lake Park, Pearl River Park, and Yunxi Ecological Park was conducted to gain insights into users' assessment of landscape services across different parks.

In terms of importance evaluation (Figure 4), safety service emerged as the most crucial aspect for park users. Firstly, all three parks' visitors assigned the highest importance to safety while attaching relatively lower significance to spiritual value, food production, and cultural preservation. Secondly, when comparing the importance levels among four types of landscape services, supply services were deemed less important compared to habitat services and regulation services. Notably, there were variations in perceptions regarding cultural services; inspiration, recreation and entertainment, physical and mental health, and safety received more attention from users, whereas spiritual value did not garner significant focus due to their subjective nature. The focus of managers should be directed towards services with higher priority.

Regarding satisfaction evaluation (Figure 5), park users expressed the highest satisfaction with physical and mental health or microclimate regulation services. Firstly, visitors at Haizhu Lake Park and Pearl River Park reported greater satisfaction with physical and mental health services, while those at Yunxi Ecological Park were most satisfied with microclimate regulation. However, all three urban parks exhibited dissatisfaction towards landscape services related to spiritual value, food production, spiritual value, and raw material supply. Secondly, differences were observed among the three urban parks concerning psycho-physical health, safety noise reduction, and nutrient cycling, but no significant differences existed for other landscape services. The landscape service with higher satisfaction is an advantageous feature of the park, and its consistent performance should be maintained. Conversely, greater attention should be given by the manager to services with lower satisfaction levels.

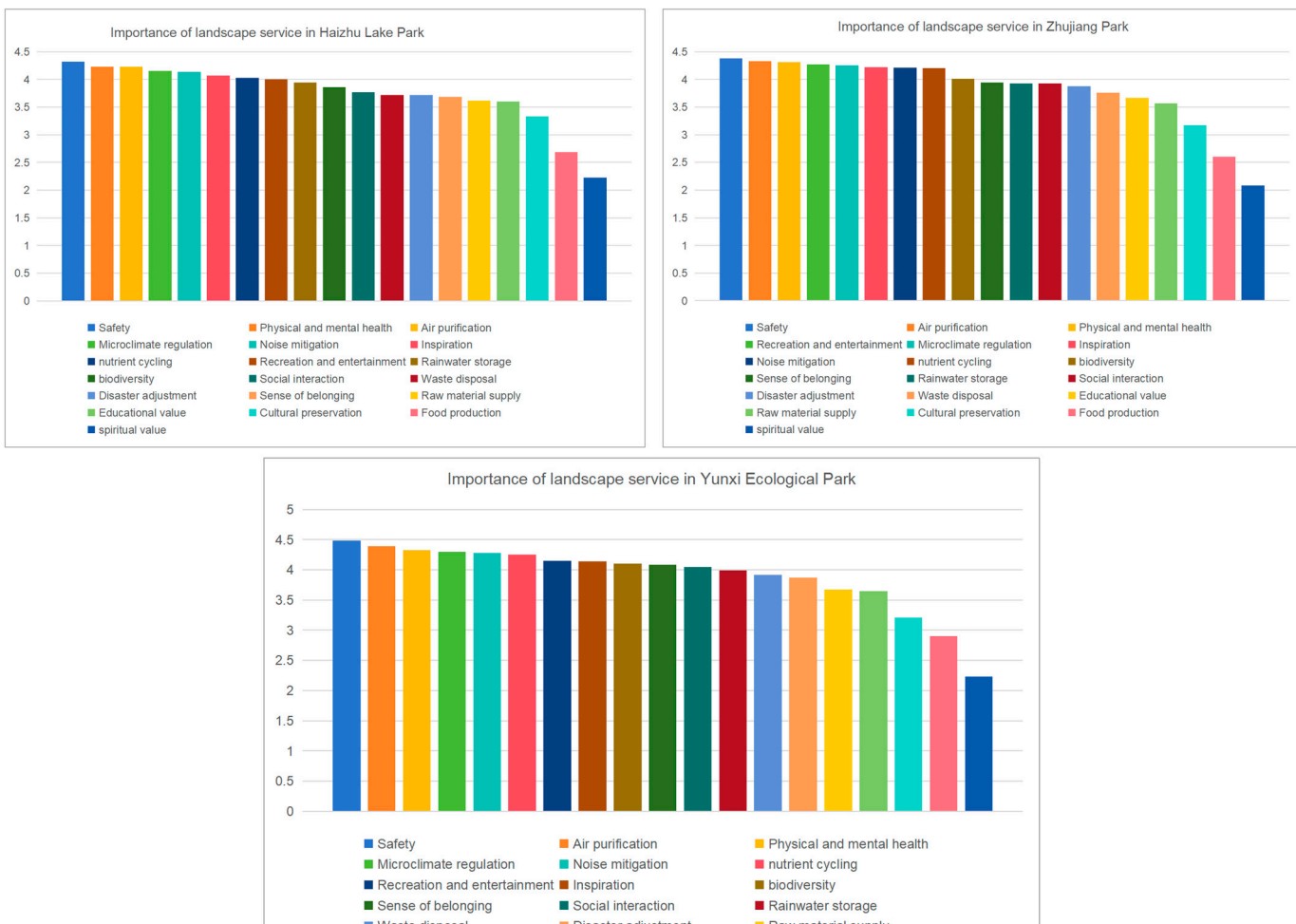

**Figure 4.** Importance ranking of urban park landscape services.

### 3.1.2. Analysis of the Differences in Perception of Landscape Service in Urban Parks

The significance and satisfaction of landscape services in the three parks were assessed using a paired-samples *t*-test. Landscape service importance was denoted as I, while landscape service satisfaction was denoted as S, with the difference between them defined as S-I. Subsequently, a paired-samples *t*-test was conducted to determine if there existed a statistically significant disparity between the two variables.

The analysis of Haizhu Lake (Table S4) reveals that only the S-I values of waste disposal, educational value, cultural preservation, and spiritual value exhibit negativity, indicating a lack of satisfaction among users regarding these aspects. Conversely, the S-I values for other services demonstrate positivity to some extent, suggesting a certain level of user satisfaction. Regarding the paired-samples *t*-test analysis, it is observed that $p = 0.376 > 0.05$ for food production and $p = 0.108 > 0.05$ for spiritual value; thus, no significant difference is found in users' perception of these two services. However, for the remaining 17 landscape services, $p = 0.000 < 0.05$; hence, there exists a significant difference in users' perception towards these services.

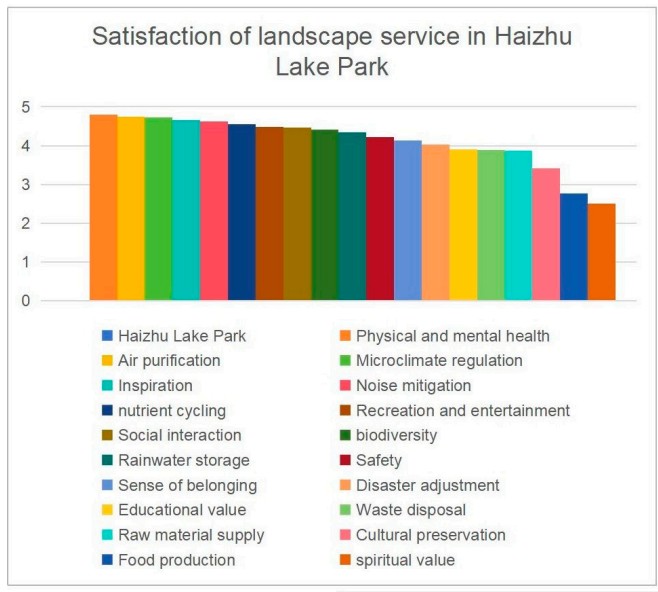

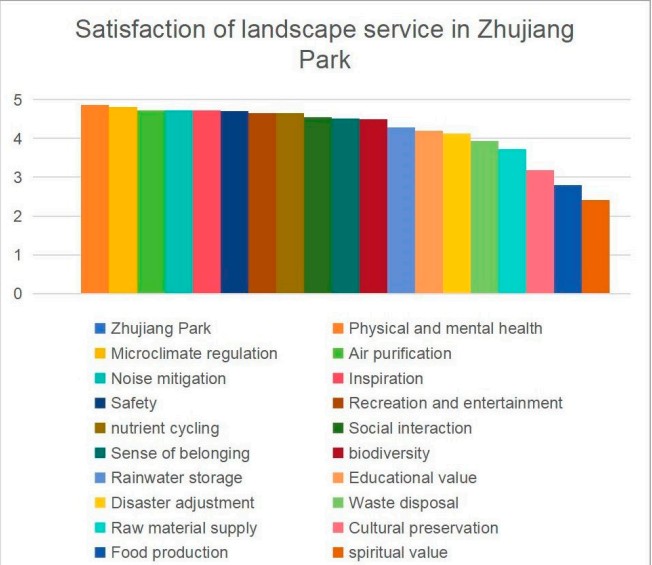

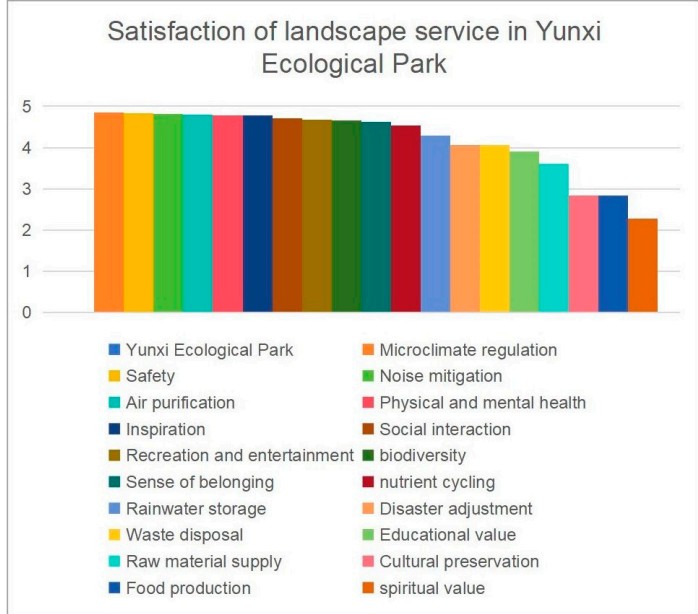

**Figure 5.** Evaluation ranking of urban park landscape service satisfaction.

The analysis of Pearl River Park (Table S5) reveals that all 19 landscape services have positive and small S-I values, indicating a certain level of satisfaction among users. Paired-samples *t*-test results indicate that there is no significant difference in users' perception of the spiritual value service ($p = 0.401 > 0.05$). However, for the remaining 18 landscape services, there is a significant difference in users' perception ($p = 0.000 < 0.05$).

Regarding the analysis of Yunxi Ecological Park (Table S5), only the S-I values for food production, raw material supply, and spiritual value exhibit negative and marginal results, indicating insufficient satisfaction of users' needs. Conversely, the S-I values for other services are positive but modest, suggesting a partial fulfillment of users' requirements. In the paired-samples *t*-test, food production ($p = 0.673 > 0.05$), raw material supply ($p = 0.478 > 0.05$), and cultural preservation ($p = 0.154 > 0.05$) demonstrate no significant disparity in users' perception towards these three landscape services; however, for the remaining 16 landscape services ($p = 0.000 < 0.05$), there exists a noteworthy distinction in users' perception.

### 3.2. IPA Analysis and Improvement Suggestions for Multifunctional Landscape of Urban Parks

The IPA method is employed for optimizing multifunctional landscapes, and an IPA optimization model of multifunctional landscapes in urban parks is developed with landscape service perception as the evaluation criterion. The quadrants of landscape services are distinguished, and the performance status and transformation priorities are analyzed to provide a basis and recommendations for enhancing the optimization of multifunctional landscapes in urban parks.

The results of IPA analysis on landscape service evaluation of the three parks are presented in the figures below (Figures 6–8). Quadrant I emerge as the dominant area encompassing safety, physical and mental health, air purification, microclimate regulation, inspiration, nutrient cycling, recreation and entertainment, rainwater storage, biodiversity, and social interaction within Haizhu Lake Park. Similarly, Pearl River Park and Yunxi Ecological Park exhibit a dominance of landscape services in Quadrant I, including safety, physical and mental health aspects along with air purification, microclimate regulation, noise reduction, inspiration, nutrient cycling, recreation and entertainment, rainwater storage biodiversity, social interaction, and sense of belonging. These identified landscape services represent key strengths that should be preserved and enhanced to fully leverage the advantages offered by each park.

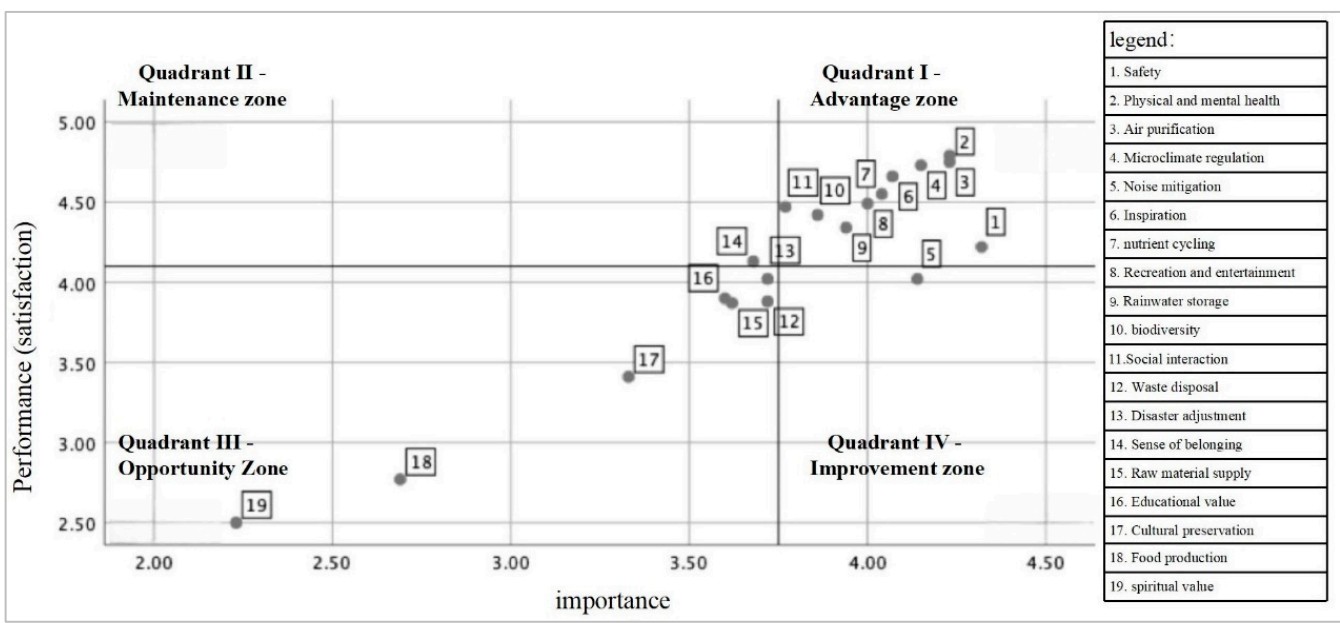

**Figure 6.** IPA analysis of Haizhu Lake Park Landscape Services.

The second quadrant represents the maintenance area, which includes the sense of belonging services in Haizhu Lake Park.

The third quadrant signifies the opportunity area, encompassing various services such as waste disposal, disaster adjustment, raw material supply, educational value, cultural preservation, food production, and spiritual value in Haizhu Lake Park; as well as waste disposal, raw material supply, educational value, cultural preservation, food production and spiritual value services in Pearl River Park; and finally, disaster adjustment service in Yunxi Ecological Park. These landscape services are prioritized for improvement to enhance users' satisfaction.

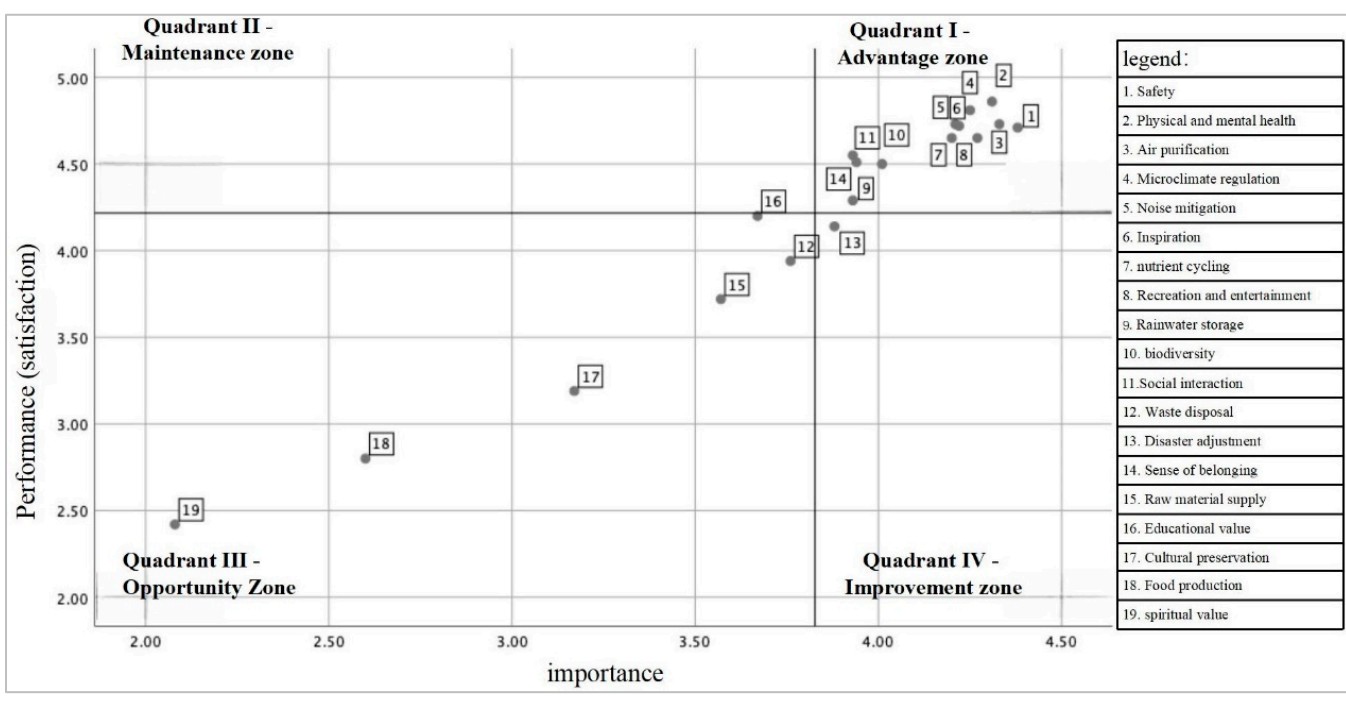

**Figure 7.** IPA analysis of landscape services in Zhujiang Park.

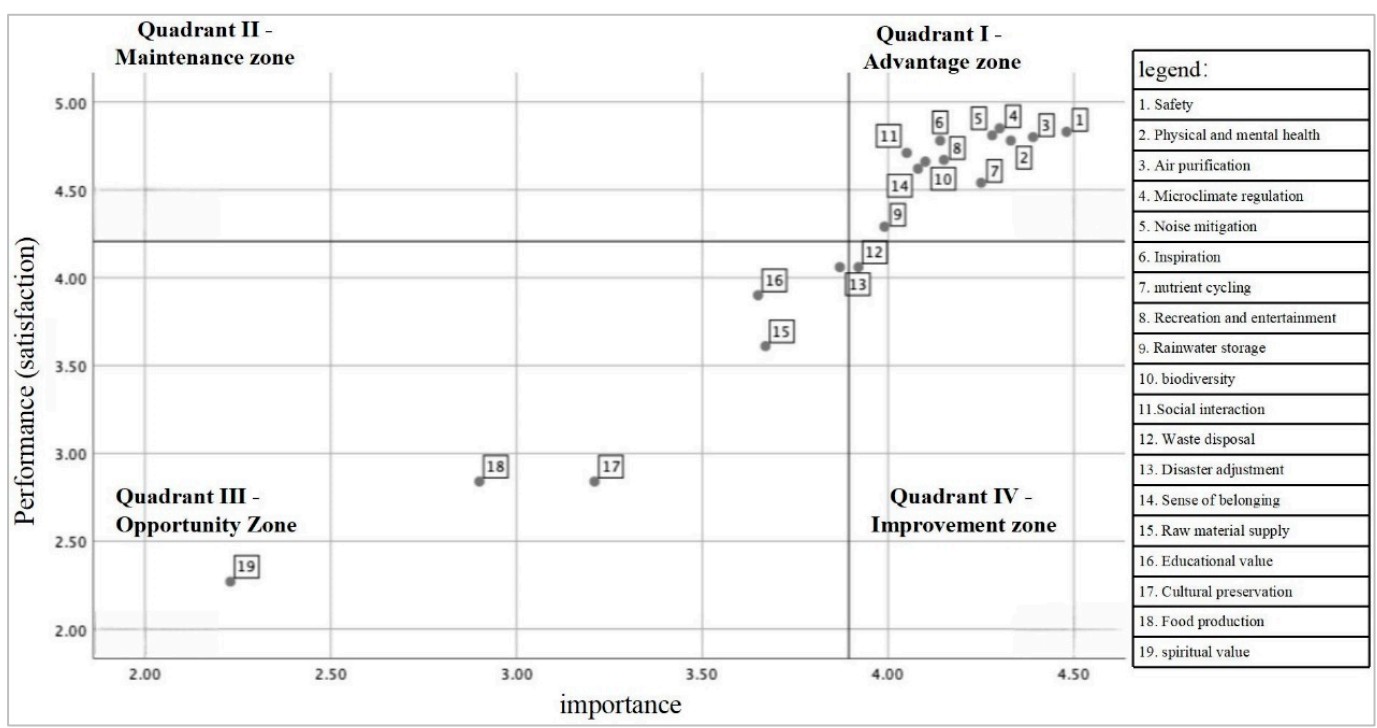

**Figure 8.** IPA analysis of landscape services in Yunxi Ecological Park.

The fourth quadrant denotes the improvement area where noise reduction service in Haizhu Lake Park is located along with disaster adjustment service in Pearl River Park and Waste disposal service in Yunxi Ecological Park. This indicates that users attach great importance to these services but remain unsatisfied, thus necessitating improvements. Table 4 presents suggestions for enhancing landscape services. The landscape service improvement proposals are presented in Table 4.

**Table 4.** Key service types and improvement suggestions for urban parks.

| Service Type | Quadrant | | | Promotion Proposal |
|---|---|---|---|---|
| | Haizhu Lake Park | Zhujiang Park | Yunxi Ecological Park | |
| Waste disposal | Opportunity zone | Opportunity zone | Improvement zone | 1. The waste classification treatment involves the fermentation of organic-rich waste into compost, thereby enhancing soil quality.<br>2. Public awareness campaigns should be conducted in parks to educate visitors about the importance of waste management and encourage their active participation in Waste disposal services. |
| disaster adjustment | Opportunity zone | Improvement zone | Opportunity zone | 1. The building should undergo lightning protection treatment and reinforcement for wooden structures.<br>2. Dead branches of trees should be promptly trimmed, and visitors should be alerted.<br>3. Emergency shelters and facilities need to be added.<br>4. Users should be guided to understand potential natural disasters and corresponding preventive measures. |
| Supply of raw materials | Opportunity zone | Opportunity zone | Opportunity zone | 1. Enhance the diversity and abundance of medicinal plants and floral species;<br>2. Disseminate ecological knowledge to park visitors through the installation of informative signage, promotional boards, as well as QR code scanning. |
| Knowledge and Education | Opportunity zone | Opportunity zone | Opportunity zone | 1. The park employs various methods, such as signage, display boards, and QR codes, to disseminate scientific knowledge about native plants.<br>2. The park facilitates the observation and promotion of common animal species.<br>3. Exhibition activities are conducted to educate the public about the diverse biological species and ecological processes within the park. |
| Spiritual value | Opportunity zone | opportunity | Opportunity zone | 1. Explore the distinctive regional cultural characteristics of Guangzhou, encompassing its rich historical and contemporary heritage.<br>2. Enhance the urban landscape by incorporating meticulously designed parks that showcase Guangzhou's captivating history and vibrant culture through interactive exhibitions.<br>3. Organize and promote traditional folk activities to foster a deeper appreciation for Guangzhou's cultural traditions. |
| food production | Opportunity zone | opportunity zone | Opportunity zone | 1. Delimit suitable areas for cultivating diverse food trees to enhance the variety and quantity of available crops;<br>2. Establish informative signage and indicators to communicate optimal seasons for planting and harvesting, thereby guiding users in experiencing the service of fruit picking. |
| spiritual religion | Opportunity zone | Opportunity zone | Opportunity zone | City park managers should exercise caution when enhancing services and thoroughly investigate the needs of users. |
| Noise Reduction | Improvement zone | Advantageous zone | Advantageous zone | 1. Continue to uphold the existing state of service provision.<br>2. Strategically position tall plants and install noise barriers in proximity to traffic routes.<br>3. Augment vegetation levels at frequently visited scenic locations.<br>4. Establish signs displaying sound decibel measurements. |

*3.3. Perception Analysis of Landscape Service in Urban Parks among Different Populations*

Due to variations in cognitive and aesthetic preferences, the landscape services valued by the public may differ significantly from those valued by planners and managers [83]. Furthermore, there are also disparities in demands for landscape services among different groups. To optimize multifunctional landscapes, it is imperative to conduct a thorough analysis of various groups' preferences for landscape services to achieve synergistic outcomes. The second-order clustering method and IPA method can be utilized to further explore diverse groups' demand preferences for landscape services.

Taking 209 respondents in Haizhu Lake Park as a case study, the second-order clustering method of SPSS 25.0 was employed to classify the respondents based on three influencing factors: age, frequency of park visits, and time spent in the park. This analysis resulted in the identification of two distinct groups with significant differences (Table 5). Among these factors, age exhibited the highest importance value of 1.0, followed by frequency of park visits with an importance value of 0.03. The least influential factor was found to be time spent in the park, with an importance value of 0.01. Therefore, age emerged as the key determinant for categorizing individuals into these two types.

**Table 5.** Analysis of Haizhu Lake Park using human clusters.

| Serial Number | Age | Frequency | Duration | Description |
|---|---|---|---|---|
| Middle-aged and elderly people | Middle age, middle age, old age (Over 30 years old) | More than 3–4 times a week, 1–2 times a week, 1–2 times a month | 2–4 h | Medium–high frequency, medium–long stay, middle-aged and elderly |
| Youth group | Youth (16–29 years old) | Every few months or less | Within 2 h or 4–8 h | Low frequency, low or high duration stay, youth |

The landscape services for the two groups were identified as quadrants through IPA analysis (Figures 9 and 10). According to the analysis, safety, physical and mental health, air purification, microclimate regulation, noise reduction, inspiration, nutrient cycle, recreation and entertainment, rainwater storage, and biodiversity are situated in advantageous areas. Haizhu Lake Park should fully leverage its advantageous services. Disaster regulation, raw material supply, educational value, cultural preservation, food production, and spiritual value fall within the opportunity area. Furthermore, the following significant disparities exist between the two groups regarding social interaction perception as well as waste disposal and sense of belonging services:

(1)	In terms of social interaction services, middle-aged and elderly individuals exhibit a lower emphasis on social interaction in the park but express a high level of satisfaction. Conversely, young people place greater importance on this service and demonstrate higher levels of satisfaction compared to the average population, indicating perceptual differences among age groups.

(2)	Regarding waste disposal, young individuals perceive it as a crucial service requiring prioritized improvement. On the other hand, middle-aged and elderly individuals consider enhancing this service as an opportunity to enhance overall services, highlighting divergent priorities for improvement.

(3)	Concerning the sense of belonging, middle-aged and elderly individuals deem the current state satisfactory. However, younger individuals believe that improvements are necessary to enhance this service.

Therefore, there exist disparities in the perception of landscape services between younger individuals and middle-aged and elderly populations. Given the aging trend within Chinese society, it is imperative to prioritize the specific needs of middle-aged and elderly individuals regarding landscape services while also guiding the optimization of such services for an aging population.

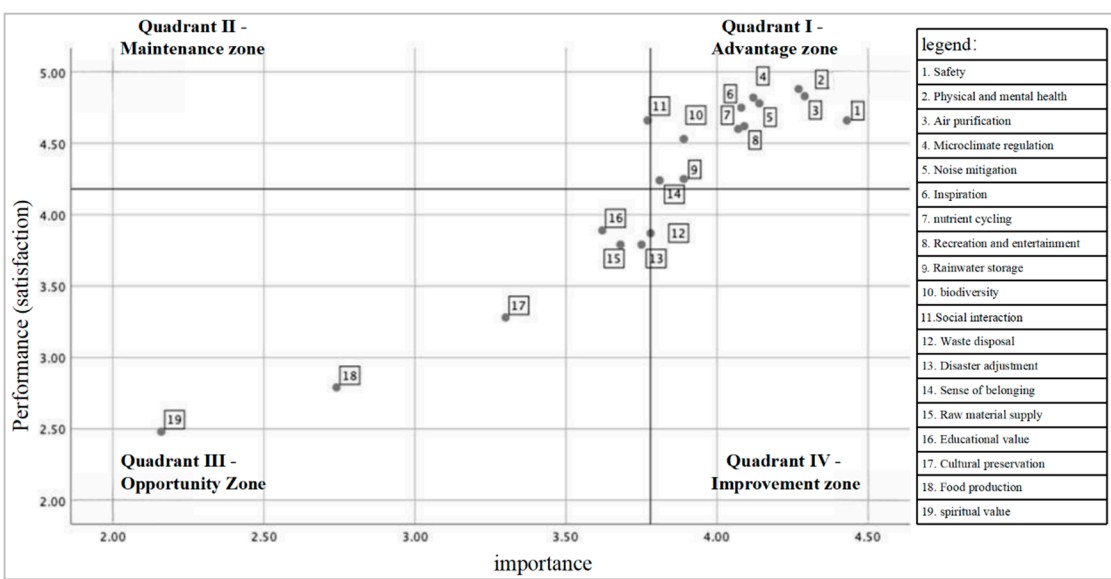

**Figure 9.** IPA Analysis of Landscape Services in Haizhu Lake Park (middle-aged and elderly people).

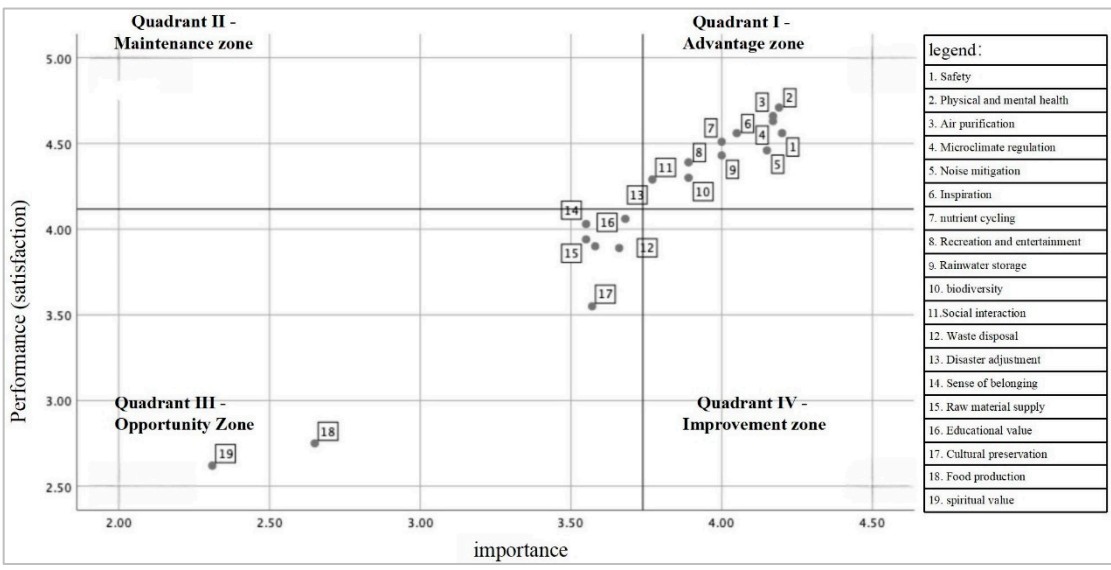

**Figure 10.** IPA Analysis of Haizhu Lake Park Landscape Services (youth population).

## 4. Discussion

### 4.1. The Public Perception of Landscape Services Facilitates Decision Making for Optimizing Multifunctional Landscapes

Firstly, this study revealed variations in citizens' perceptions of different landscape services, with a particular emphasis on security services. This highlights citizens' heightened concern for the safety of urban parks, especially considering their integration into urban life. This may be attributed to the prevailing threat of crime within parks, as supported by relevant studies. For instance, Maruthaveeran posited that individuals are more inclined to visit parks accompanied by family and friends rather than alone [84]; Sara et al. research demonstrated citizens' increased apprehension regarding park-related crimes [85]; Sharifah et al. suggested that plants can facilitate criminal activities [86]. Furthermore, this could also be attributed to the presence of wildlife since these three urban parks in Guangzhou boast rich biodiversity and dense vegetation, harboring numerous potentially harmful wild animals such as snakes. Secondly, this study revealed that citizens assigned the least importance to spiritual sustenance, followed by food supply and cultural heritage.

This could be attributed to the subjective nature of preferences for spiritual sustenance and cultural heritage despite their contribution to fostering local distinctiveness and a sense of belonging. These findings may also be associated with respondents' age and educational background. Furthermore, users demonstrated less concern towards food supply compared to scholarly perspectives. Currently, academia emphasizes the edible value of urban landscapes [87], highlighting a disparity between academic viewpoints and residents' perspectives. Additionally, this study identified that citizens prioritize physical and mental health services as well as microclimate regulation services, which is in line with Maruthaveeran's findings [84]. Consequently, parks serve as spaces for individuals to inhale fresh air, alleviate stress, and unwind. Finally, this study shows that citizens are most dissatisfied with spiritual sustenance, food supply, cultural heritage, and raw material supply, and when the user's satisfaction increases, it can promote their positive behavior [88]; therefore, although there is fruit picking function in Haizhu Lake Park and Yunxi Ecological Park, residents' dissatisfaction indicates that this service needs to be improved, and there is no fruit picking service in Pearl River Park, so this service needs to be added.

Meanwhile, the paired-samples *t*-test further substantiated the disparities in landscape service perception. Although subjective to some extent, such evaluation reflects the public's demand and preference for landscape services and can provide valuable information for optimizing multifunctional landscapes. Unlike traditional landscape planning, multifunctional landscape planning and management places greater emphasis on stakeholder input [4]. Furthermore, taking into account specific stakeholders' demands for landscapes and clarifying construction directions of multifunctional landscapes can enhance ecosystem services and human well-being, which has gradually become a policy practice at local levels [89]. Wu's research also confirmed that urban park users' satisfaction helps public managers improve their provision of landscape services [90]. Therefore, in the future, urban park managers should optimize decision making regarding those aspects of landscape services that are highly valued by but have not yet satisfied the public so as to better construct multifunctional landscapes that meet their needs. Additionally, analyzing public perceptions can help distinguish differences between planners/managers' values regarding landscape services from those held by the general public so as to balance personal interests with broader societal goals.

### 4.2. IPA Analysis Can Optimize Multifunctional Landscapes

This study demonstrates that the IPA analysis method effectively identifies the prioritization of multifunctional landscape optimization. Specifically, in Haizhu Lake Park, the foremost service requiring improvement is noise reduction. Although park users highly value tranquility, they express dissatisfaction due to noise pollution caused by a nearby motorway, which hinders their ability to enjoy a peaceful recreational environment. Previous research has also highlighted the adverse effects of urban park noise on users [91]. Additionally, waste disposal, disaster adjustment, raw material supply, knowledge education, cultural preservation, food provision, and spiritual value are secondary services that necessitate enhancement. While park users currently do not prioritize or find satisfaction in these services individually, improving them can optimize the overall quality of the park's multifunctionality. Arijit Das et al. also demonstrated that urban residents exhibit limited attention towards and dissatisfaction with knowledge education and food supply [55]. Therefore, the management of Haizhu Lake Park should prioritize waste treatment and enhance waste recycling efforts. Additionally, the cultivation of certain traditional Chinese medicine materials can be considered for raw material supply while promoting natural education and knowledge services to enrich the park's cultural heritage and traditional festival activities. Other services may simply require maintenance to sustain their current status.

This study revealed that the disaster adjustment service in Pearl River Park is of utmost urgency. Previous research has established a correlation between residents' demands and disasters [92]. This could be attributed to the frequent occurrence of lightning and

typhoon weather in Guangzhou, coupled with inadequate protective measures within the park, leading to visitor dissatisfaction with the service. Furthermore, during the summer rainy season, the lack of fluidity in the park's lake can result in waterlogging, negatively impacting residents' experiences. Consequently, park management should allocate more resources towards enhancing service quality. Waste disposal, raw material supply, knowledge education, cultural preservation, food provision, and spiritual value constitute secondary priority services. In Yunxi Ecological Park, waste disposal is the primary service requiring improvement. However, the park demonstrates efficient and organized garbage disposal with timely collection and proper storage facilities, particularly for greening waste. Remarkably, this practice has significantly enhanced soil fertility. The deficiency in effective publicity may account for the lack of public awareness and acceptance regarding the park's greening waste management approach.

The emergence of multifunctionality does not stem from deliberate planning but rather arises from the intricate interactions among diverse actors with distinct goals and needs [93,94]. Effectively prioritizing the updating of different attributes poses a formidable challenge for global urban governance [42]. Through IPA analysis, this study identifies priority indicators for various landscape service updates based on public importance and satisfaction evaluation. It further prioritizes the optimization of landscape services in both opportunity areas and improvement areas, determining the sequence for multifunctional landscape optimization. This approach offers practical guidance for achieving multifunctional landscape optimization.

### 4.3. Different Groups Have Different Demands for Landscape Services

This study examined variations in demand for landscape services across different demographic groups. It identified two distinct user groups, namely the young and the middle-aged and elderly, which exhibited significant disparities in their preferences for social interaction, waste disposal, and sense of belonging. This finding further corroborated previous research indicating that there are social group discrepancies in the perception of ecosystem services [95,96]. For instance, Shijie Gai et al. also affirmed that users' age and residence can influence their perception of ecosystem services [59]. The middle-aged and elderly group prioritized waste treatment improvements more than the young group did; however, they placed less emphasis on enhancing place-sense. This discrepancy may be attributed to the middle-aged and elderly group's greater sense of belonging in Haizhu Lake Park or could be associated with their longer duration of residency in Guangzhou. Additionally, both user groups identified disaster adjustment, raw material supply, knowledge education, cultural preservation, food production as well as spiritual value as key landscape services requiring priority upgrades.

In previous environmental renewal practices, the input of communities and residents was rarely sought [18]. During the process of environmental planning and decision making, various landscapes were often converted into land use types with homogeneous structures and singular functions [97,98], leading to conflicts and contradictions among different stakeholders' demands. Therefore, when formulating policies for optimizing multifunctional landscapes in urban parks, it is crucial to fully consider the differences between user groups and adopt diverse renewal strategies, particularly addressing the specific needs of elderly individuals in an aging society. Relevant studies also demonstrate that urban parks serve as vital public service facilities with significant implications for maintaining physical and mental well-being among older adults while enhancing economic, social, and environmental benefits within cities [99–101].

### 5. Conclusions

The IPA model of multifunctional landscape optimization developed in this study has made significant technological and practical contributions. Firstly, it integrates stakeholder needs into the decision-making process for managing multifunctional landscapes. This study employs the concept of landscape services to establish an evaluation index system for

urban park landscapes, bridging the gap between public perception and multifunctional landscapes. By employing a Likert scale questionnaire, it quantifies public importance and satisfaction with landscape services, enabling a comprehensive understanding of their demands and facilitating evaluative assessments of their value. This approach addresses the limitations of previous park renewal efforts that relied solely on expert decision making by providing valuable insights into the diverse needs of park users, thereby offering essential information for optimizing management decisions regarding multifunctional landscapes in urban parks. Furthermore, the IPA method offers a systematic approach to prioritize multifunctional landscape renovation. Multifunctional landscapes represent an optimal stage of development, encompassing diverse functions such as ecological, economic, cultural, historical, and aesthetic aspects [102]. In the context of managing urban park landscapes with multiple functions, determining which services should be updated first poses a challenging problem in terms of cost-effectiveness and resource efficiency. The IPA method facilitates the transformation of importance and satisfaction evaluations into priority indicators, enabling managers to identify key issues and allocate resources efficiently.

## 5.1. Implications for Public Decision Makers

The evaluation of the significance and satisfaction of landscape services for urban park users in this study holds practical implications for optimizing multifunctional landscape design and formulating management strategies for urban parks.

Firstly, this study can directly provide managers with recommendations for optimizing the multifunctional landscape of parks. This study unveils the prioritization of multifunctional landscape optimization in three urban parks. In Haizhu Lake Park, the primary focus lies in enhancing noise reduction services, such as implementing tall vegetation to create a buffer zone between the park and surrounding motorways, mitigating the impact of vehicular noise on park visitors. Additionally, installing sound decibel signage can guide visitors away from noisy areas. In Pearl River Park, priority is given to improving disaster resilience services through measures like structural reinforcement and lightning protection systems installation, establishing emergency shelters, and conducting disaster prevention awareness campaigns and drills. Lastly, Yunxi Ecological Park emphasizes enhancing waste management services encompassing garbage sorting and treatment practices along with regular cleaning operations while fostering public participation; furthermore, increasing the number of garbage bins is recommended. In light of the deficiencies in knowledge education and cultural history within Guangzhou parks, it is imperative for policymakers to integrate nature education and historical/cultural elements into park landscape functions through comprehensive planning, design, effective publicity, and active public participation. However, there are certain obstacles that hinder the improvement of existing parks based on the findings of this study. The present research did not investigate the relationship between landscape services and specific types or spaces within parks, potentially resulting in some blind spots when implementing relevant strategies. In future studies, it would be beneficial to spatially consider residents' perception of landscape services; for example, the questionnaire incorporates landscape element images with geographical coordinates for evaluation purposes. Additionally, this study did not conduct in-depth interviews regarding the specific reasons behind park users' dissatisfaction with landscape services; this may also impact the effectiveness of multifunctional landscape optimization strategies. Future research should aim to refine our understanding by exploring residents' perceptions more comprehensively. For instance, when perception is refined into distinct modalities such as hearing, vision, smell, touch, and others [60], it can be further associated with various landscape elements.

Secondly, this study also prompted park managers and planners to critically evaluate the top-down approach to optimizing multifunctional landscapes. It is crucial for decision makers in parks to acknowledge that urban parks primarily serve ordinary citizens rather than solely government officials. Failure to recognize this can lead to a disconnect between

park design and usage, resulting in suboptimal landscape services for citizens. Previous research has highlighted disparities between decision makers' perceptions and residents' needs [103,104]. This study emphasizes the importance of prioritizing multifunctional landscape optimization based on public opinions as expressed by residents, offering valuable guidance and suggestions for enhancing urban park landscapes. Therefore, future efforts toward optimizing multifunctional landscapes should involve mechanisms and strategies that facilitate public participation, including gathering resident feedback through interviews, surveys, and planning hearings.

Thirdly, this study unveils the intricate correlation between urban dwellers and their demand for landscape services, enabling park policymakers to devise landscape optimization strategies tailored to specific demographic groups. Previous studies by Swapan et al. [57] and Dou et al. [105] have also substantiated those factors such as age and gender influence residents' preferences for landscape services. Hua and Chen [96] discovered that young individuals place a greater emphasis on the social aspects of parks. The findings of this study expose disparities in perceptions between middle-aged/elderly individuals and younger counterparts. Apart from age, residents' income, educational background, and proximity to parks can impact their preference for landscape services. Consequently, when implementing multifunctional landscape optimization in urban parks, managers should further discern the diverse needs of different groups in order to create versatile landscapes capable of satisfying varying demands.

*5.2. Research Limitations and Prospects*

This method also has certain limitations that can be addressed in future research. Firstly, IPA analysis exhibits evident collinearity due to the data's origin from questionnaire surveys, where respondents seldom provide negative evaluations on survey questions. Additionally, the sample size of the middle-aged and elderly group is relatively small, with the middle-aged and elderly (50–65) comprising only 15.3% and those aged over 75 accounting for merely 3.0%. Consequently, this weakens the evaluation of park landscape services by older adults and fails to fully reflect their actual perception. Therefore, future studies should consider incorporating multi-source data. For instance, mobile phone signaling data and interview data are utilized in this study. Mobile phone signaling data encompasses demographic attributes such as age, gender, and movement patterns. Secondly, respondents lacking a sufficiently accurate understanding of landscape services may introduce their own subjective speculations during the evaluation, thereby influencing the analysis results to some extent. For instance, the nutrient cycle is simplified as "the park exhibits high oxygen content and low carbon dioxide content during the day." However, the nutrient cycle also encompasses essential elements such as nitrogen and sulfur. Nevertheless, from a user perception standpoint, evaluating oxygen and carbon dioxide levels appears more evident and convenient, resulting in an incomplete representation of the nutrient cycle. Consequently, future research should incorporate more detailed categorization and formulation of perception questions to enhance data accuracy and reliability. Thirdly, this study solely focused on the impact of age on population division, neglecting to consider other influential factors, such as the spatial attributes of the park itself on residents [106,107]. Therefore, future research should expand upon these influencing factors and compare their interrelationships in order to uncover and address diverse populations' landscape needs and preferences more comprehensively, ultimately shaping multifunctional landscapes. Fourthly, the absence of spatialization in this study undermines the accuracy of the relationship between landscape services and spatial types and elements, thus necessitating future studies to incorporate a spatial perspective when assessing residents' perception of landscape services.

**Supplementary Materials:** The following supporting information can be downloaded at: https://www.mdpi.com/article/10.3390/land13050564/s1, Table S1: Demographic characteristics of the sample; Table S2: Reliability test; Table S3: Validity test; Table S4: Paired sample *t*-test of importance

and satisfaction of Haizhu Lake Park; Table S5: Paired sample *t*-test of importance and satisfaction in Zhujiang Park; Table S6: Paired sample *t*-test of importance and satisfaction of Yunxi Ecological Park.

**Author Contributions:** Conceptualization, X.X.; methodology, X.X. and X.D.; software, X.X. and Q.Y.; validation, X.X.; formal analysis, X.X. and Q.Y.; investigation, X.X. and X.D.; resources, X.X.; data curation, X.X. and X.D.; writing—original draft preparation, X.X., X.D. and Q.Y.; writing—review and editing, X.X. and Q.Y.; visualization, X.X. and Q.Y.; project administration, X.D. All authors have read and agreed to the published version of the manuscript.

**Funding:** This research was funded by the National Natural Science Foundation of China, grant number 42171275, and the Second Tibetan Plateau Scientific Expedition and Research (STEP) program, grant number 2019QZKK0608; KJZXCJ2019366, and the specific research fund of the Innovation Platform for Academicians of Hainan Province, grant number YSPTZX202308.

**Data Availability Statement:** Data are contained within the article.

**Conflicts of Interest:** The authors declare no conflicts of interest.

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
