# Peer review of "Using Importance–Performance Analysis to Reveal Priorities for Multifunctional Landscape Optimization in Urban Parks"

_land, doi:10.3390/land13050564_

Round 1

Reviewer 1 Report

Comments and Suggestions for Authors

This manuscript presents a study that addresses the challenge of optimizing the multifunctional landscape of urban parks based on residents' perceptions of landscape services. The authors integrate Importance-Performance Analysis (IPA) with a perception evaluation of landscape services to establish an index system for assessing urban park landscape services. It uses Guangzhou's urban parks as a case study, revealing variations in user demands for different landscape services and identifying priorities for landscape renewal.

1. Literature Review: The literature review is well-structured and covers the necessary theoretical background. However, it could benefit from a more critical analysis of the existing literature, highlighting specific gaps and how the current study addresses them.

2. Research Questions and Identifying Gaps: The study poses relevant research questions regarding the optimization of multifunctional landscapes in urban parks and the role of IPA in identifying priorities for improvement. The authors effectively identify gaps in the current understanding of how to integrate stakeholder needs into the planning and management of multifunctional landscapes. However, the manuscript could be improved by providing a more detailed discussion on how these gaps can be addressed through future research, including potential research questions that could guide such studies.

3. Research Methods and Models: The methodology section of the paper is robust, detailing the use of IPA in conjunction with a questionnaire to evaluate perceptions of urban park landscape services. The authors describe the process of data acquisition and analysis, including the use of statistical tools such as Cronbach's Alpha and paired sample T-tests. The IPA model is well-explained, and the authors demonstrate its application in the context of urban park landscape optimization. However, the manuscript could be strengthened by providing more specific examples of how the IPA model has been applied in other contexts and the challenges encountered in doing so.

4. Results and Discussion: The results of the study reveal significant findings regarding user perceptions of landscape services in urban parks and highlight the potential of IPA analysis in prioritizing landscape renewal efforts. The authors discuss the implications of their findings for urban park landscape management and provide actionable recommendations for improvement. The discussion section could be enhanced by a more in-depth analysis of the implications for policy and practice, including potential barriers to implementation.

Overall, the manuscript makes a valuable contribution to the field of urban park landscape optimization, there are areas for improvement. The authors could enhance the paper by:

  1. Providing a more critical analysis of the literature, identifying specific gaps and how they limit our understanding of multifunctional landscapes.
  2. Offering a more detailed discussion on the potential research questions that could address the identified gaps.
  3. Including case studies or examples that demonstrate the application of the IPA model in real-world scenarios and the challenges encountered.
  4. Delving deeper into the implications of the research findings for policy and practice, including potential barriers to implementation and strategies for overcoming them.

Author Response

Thank you for taking the time and effort to review my manuscript and revise it according to your suggestions. Please see the attachment.

Reviewer 2 Report

Comments and Suggestions for Authors

This study uses the Importance-Performance Analysis (IPA) with residents' perception to evaluate the landscape services of three urban parks in Guangzhou, China. The research topic is relevant and the paper provides a valuable contribution to the field, therefore, I recommend the acceptance of the paper after a minor revision. 

Authors must update the literature review with the following references:

Das, A., & Basu, T. (2020). Assessment of peri-urban wetland ecological degradation through importance-performance analysis (IPA): A study on Chatra Wetland, India. Ecological Indicators114, 106274.

Wang, Z., Miao, Y., Xu, M., Zhu, Z., Qureshi, S., & Chang, Q. (2021). Revealing the differences of urban parks’ services to human wellbeing based upon social media data. Urban Forestry & Urban Greening63, 127233.

Kong, L., Liu, Z., Pan, X., Wang, Y., Guo, X., & Wu, J. (2022). How do different types and landscape attributes of urban parks affect visitors' positive emotions?. Landscape and Urban Planning226, 104482.

Gai, S., Fu, J., Rong, X., & Dai, L. (2022). Users’ views on cultural ecosystem services of urban parks: An importance-performance analysis of a case in Beijing, China. Anthropocene37, 100323.

Zheng, T., Yan, Y., Lu, H., Pan, Q., Zhu, J., Wang, C., ... & Zhan, Y. (2020). Visitors’ perception based on five physical senses on ecosystem services of urban parks from the perspective of landsenses ecology. International Journal of Sustainable Development & World Ecology27(3), 214-223.

In the text, reference numbers should be placed in square brackets [ ], and placed before the punctuation; for example [1], [1–3] or [1,3].

The text in section 2.5.1 should inform the sample dimension of each survey collected in the study.

The Chinese words in the heading of Table 6, must be translated into English.

Authors should add a new section "5.1- Implications for public decision maker" in order to improve the discussion of the proposals of Table 11. Authors should also analyse the limitations of study and suggest further research.

Author Response

(The authors gave the same response as above.)

Reviewer 3 Report

Comments and Suggestions for Authors

The adopted research topic regarding revitalization directions is interesting and fits the Land journal profile. However, there are several errors in the presented article that should be corrected.

Generally, the article requires serious editing. The sources are given chaotically and often illegibly, contrary to the editorial requirements of Land magazine. In many places this makes it impossible to fully understand the text.

The introduction is messy. Although the chapter presents the main assumptions related to the described research, it does so in a somewhat haphazard manner. The research background requires supplementing with legal guidelines related to shaping and using the landscape at the local (Chinese) and global levels. It would be advisable to show and compare examples of similar activities from other countries.

The methodology and materials are presented quite broadly and would require additional diagrams showing the sequence of activities adopted in the study. Figure 1 is in my opinion insufficient. Tables 5, 6 and 7 should be included in appendices outside the main text. Part of Table 6 requires translation.

The results are also presented in a somewhat chaotic way and require tidying up. The statistics are presented correctly, but Tables 8, 9 and 10 should also be included in the appendices.

The discussion should refer to the main theses and results of the study, it requires extension. Conclusions should include the main recommendations related to the future development of the described landscape spaces.

Author Response

(The authors gave the same response as above.)

Round 2

Reviewer 1 Report

Comments and Suggestions for Authors

Thanks to the author for the full revision and improvement in this edition, the current draft has been greatly improved. There are a few minor issues that authors need to be aware of:

1. The study employed Importance-Performance Analysis (IPA) combined with residents' perception evaluations of landscape services to establish an index system for assessing urban park landscape services. Quantitative assessments of landscape services were made using a Likert scale survey, providing an objective basis for prioritizing landscape services for optimization. In this regard, there are several aspects that the authors may need to elaborate on: 1) Sample Representativeness: The study sample may lack sufficient representation, particularly with a smaller sample size of middle-aged and elderly individuals, which may not fully reflect this group's actual perceptions. 2) Subjectivity: The survey may be influenced by the respondents' subjectivity, leading to potential biases in the understanding of certain landscape services. 3) Single Data Source: The study primarily relied on survey data, lacking support from other data sources such as field observations or expert interviews.

2. In the presentation of results, the study did not explore the relationship between landscape services and specific types or spaces within the park, which may present blind spots when implementing relevant strategies. Meanwhile, the study did not conduct an in-depth analysis of the specific reasons behind dissatisfaction with certain landscape services, which may affect the effectiveness of multifunctional landscape optimization strategies.

3. Finally, the discussion lacked an in-depth exploration of the connection between existing theories and practices. The long-term effects and sustainability of optimization strategies were not fully discussed.

Author Response

Thank you for taking the time and effort to review my manuscript and please see the attachment.

Reviewer 3 Report

Comments and Suggestions for Authors

The article has been significantly improved. The authors' work is visible in a satisfying way. The introduction presenting the research background has been extended to include the necessary legal provisions regarding global requirements. The methodology was presented much more clearly. The results were cleared from unnecessary tables. The discussion has been extended. In general, the structure of the article has been tidied up.

Author Response

(The authors gave the same response as above.)
